# Bone Remodeling Markers in Children with Acute Lymphoblastic Leukemia after Intensive Chemotherapy: The Screenshot of a Biochemical Signature

**DOI:** 10.3390/cancers15092554

**Published:** 2023-04-29

**Authors:** Paola Muggeo, Massimo Grassi, Vito D’Ascanio, Vincenzo Brescia, Antonietta Fontana, Laura Piacente, Francesca Di Serio, Paola Giordano, Maria Felicia Faienza, Nicola Santoro

**Affiliations:** 1Department of Pediatric Oncology and Hematology, University Hospital of Policlinic, 70124 Bari, Italy; paola.muggeo@gmail.com (P.M.); grassimassimo@hotmail.it (M.G.); nico.santoro1956@libero.it (N.S.); 2Institute of Sciences of Food Production (ISPA), Italian National Research Council (CNR), 70126 Bari, Italy; vito.dascanio@ispa.cnr.it; 3Clinical Pathology Unit, AOU Policlinico Consorziale di Bari-Ospedale Giovanni XXIII, 70124 Bari, Italy; bresciavincenzo58@gmail.com (V.B.); antonietta.fontana@policlinico.ba.it (A.F.); francesca.diserio@policlinico.ba.it (F.D.S.); 4Pediatric Unit, Department of Precision and Regenerative Medicine and Ionian Area, University “A. Moro”, 70124 Bari, Italy; laura.piacente@uniba.it; 5Interdisciplinary Department of Medicine, University of Bari “Aldo Moro”, 70124 Bari, Italy

**Keywords:** acute lymphoblastic leukemia (ALL), bone remodeling, bone biomarkers, nuclear factor kappa-B ligand (RANKL), osteoprotegerin (OPG), Wnt/β-catenin, sclerostin, Dickkopf-1 (DKK-1)

## Abstract

**Simple Summary:**

Acute lymphoblastic leukemia (ALL) is the most frequent pediatric malignancy. The survival of ALL patients has reached a 90% 5-year survival rate, thanks to intensive chemotherapy regimens. Improved survival of ALL patients has led to an increase in long-term complications of chemotherapy, including adverse effects on bone, such as osteopenia/osteoporosis, osteonecrosis, and fragility fractures. Skeletal health depends on the balance between bone resorption and bone deposition, through “bone remodeling” coordinated by the nuclear factor kappa-B ligand (RANKL)/RANK/osteoprotegerin (OPG) and Wnt/β-catenin pathways, respectively. There are no data on the effect of intensive chemotherapy on bone remodeling markers in ALL children. We investigated these effects and characterized the unknown biochemical signature of bone status in these patients. Our cohort of ALL children showed a biomarker profile of increased bone resorption and ineffective bone formation. This condition can expose them to the risk of osteopenia and increased bone fragility.

**Abstract:**

Purpose: to investigate the effects of intensive chemotherapy and glucocorticoid (GC) treatment on bone remodeling markers in children with acute lymphoblastic leukemia (ALL). Methods: A cross-sectional study was carried out in 39 ALL children (aged 7.64 ± 4.47) and 49 controls (aged 8.7 ± 4.7 years). Osteoprotegerin (OPG), receptor activator of NF-κB ligand (RANKL), osteocalcin (OC), C-terminal telopeptide of type I collagen (CTX), bone alkaline phosphatase (bALP), tartrate-resistant acid phosphatase 5b (TRACP5b), procollagen type I N-terminal propeptide (P1NP), Dickkopf-1 (DKK-1), and sclerostin were assessed. Statistical analysis was conducted using the principal component analysis (PCA) to study patterns of associations in bone markers. Results: ALL patients showed significantly higher OPG, RANKL, OC, CTX, and TRACP5b than the controls (*p* ≤ 0.02). Considering ALL group, we found a strong positive correlation among OC, TRACP5b, P1NP, CTX, and PTH (r = 0.43–0.69; *p* < 0.001); between CTX and P1NP (r = 0.5; *p* = 0.001); and between P1NP and TRAcP (r = 0.63; *p* < 0.001). The PCA revealed OC, CTX, and P1NP as the main markers explaining the variability of the ALL cohort. Conclusions: Children with ALL showed a signature of bone resorption. The assessment of bone biomarkers could help identify ALL individuals who are most at risk of developing bone damage and who need preventive interventions.

## 1. Introduction

Acute lymphoblastic leukemia (ALL) is the most frequent pediatric malignancy and represents almost 1/4 of childhood cancers [1]. In the last decade, the 5-year survival rate has reached about 90% because of risk-stratified multiagent chemotherapy and the improvement of supportive care [2]. However, the long-term complications have also risen, both in terms of cardiometabolic alterations [3,4,5] and impact on skeletal mineralization [6]. A high dose of glucocorticoids (GCs) represents a milestone in the treatment of ALL. The use of dexamethasone or prednisone is crucial during the induction and delayed intensification phases, which last approximately 4 weeks each, and during the consolidation phase in high-risk patients. Glucocorticoid (GC)-induced osteoporosis (GIO) is the most common cause of secondary osteoporosis. GCs decrease bone formation by promoting apoptosis of osteoblasts and osteocytes and increasing bone resorption [7,8]. Furthermore, GCs reduce intestinal absorption of calcium and increase its renal excretion; thus, fractures may occur in 30–50% of patients on chronic GC therapy [9]. In children suffering from ALL, a reduction in bone mineral density (BMD) and osteoporosis have already been observed at the time of diagnosis, implying direct effects of cancer per se, probably due to both the bone infiltration by malignant cells and the action of autocrine factors [10,11]. Indeed, a prospective study in ALL adolescents and young adults demonstrated significant alterations of cancellous and cortical bone during the first month of treatment [12]. Bone loss has also been documented after the end of chemotherapy as a late effect in young and adult survivors of pediatric ALL [13]. In addition to bone loss, osteonecrosis represents one of the most prevalent therapy-related adverse effects in ALL children and adolescents with a long-term negative impact on quality of life [14]. GCs are primarily responsible for the development of osteonecrosis. A correlation has been observed between cumulative GC dose and the risk of osteonecrosis in subjects with ALL, especially in older patients due to slower clearance which may contribute to increased bone toxicity [15,16,17,18].

Bone remodeling is essential for bone homeostasis. It is due to a balance between the two phases of bone formation and bone resorption. This equilibrium is mostly regulated by the nuclear factor kappa-B ligand (RANKL)/RANK/osteoprotegerin (OPG) and Wnt/β-catenin pathways, which control osteoclastogenesis and osteoblastogenesis, respectively [19,20]. RANKL binds to RANK on myeloid cells and induces differentiation and activation of osteoclast precursors, leading to increased bone resorption. OPG is a soluble decoy receptor for RANKL which prevents the RANKL-RANK binding and inhibits osteoclast maturation, indicating that the RANKL/OPG ratio plays a key role in the regulation of bone remodeling. Furthermore, the pro-osteoblastogenic activity of the Wnt/β-catenin pathway can be inhibited by sclerostin and Dickkopf-1 (DKK-1). High serum levels of RANKL, sclerostin, and DKK-1 have been found in several pediatric bone diseases [21]. Furthermore, these bone cytokines represent the target of new treatments for osteopenia/osteoporosis [21].

Recently, it has been demonstrated that GCs modulate the bone remodeling markers in subjects affected by rheumatoid arthritis (RA), and they are responsible for the reduction in BMD and alteration in bone quality depending on dose regimens [22]. There are no data about bone remodeling markers in pediatric patients treated with high-dose GC and intensive chemotherapy for ALL.

The aim of this study was to investigate bone remodeling biomarkers after intensive phases of chemotherapy in a cohort of ALL children and to characterize the unknown biochemical signature of bone status in these patients. For this purpose, we assessed the serum levels of RANKL, OPG, sclerostin, DKK-1, and bone metabolism markers, and we evaluated the correlations between bone remodeling cytokines, bone metabolism markers, and clinical data. A better understanding of bone remodeling balance could enhance the prevention of osteopenia/osteoporosis in these patients.

## 2. Materials and Methods

### 2.1. Subjects

The study group consisted of 39 children diagnosed with ALL (21 males), mean age at recruitment 7.6 ± 4.4 years, treated according to the ongoing international AIEOP-BFM ALL protocols. Patients were recruited between June 2020 and June 2022, at the Pediatric Hematology and Oncology Clinic, University Hospital of Bari, Bari, Italy. All the patients had ended the intensive phases of chemotherapy, namely the induction, consolidation, and re-induction (or delayed intensification) phases, and were receiving maintenance treatment (21 patients) or had ended chemotherapy (18 patients). Inclusion criteria were (a) age 3 to 20 years at the recruitment, corresponding to the age of 1–18 years at the diagnosis of ALL; (b) diagnosis of ALL; (c) end of intensive phases of chemotherapy; and (d) complete remission of ALL.

The control group consisted of 49 healthy subjects (21 males), pair matched by age (8.7 ± 4.7 years), attending the Pediatric Clinic of the University of Bari for minor trauma (first aid) or allergology screening. Exclusion criteria from this study for both patients and controls were the use of vitamin and mineral supplements, the presence of chronic diseases with a possible impact on bone metabolism (e.g., hypothyroidism or hyperthyroidism, Cushing’s syndrome, celiac disease, anorexia nervosa), genetic syndromes, and fractures in the 6 months preceding the study.

The study protocol was approved by the Local Ethic Committee. Written informed consent was signed by both parents or by patients aged older than 18 years. All the procedures used were in accordance with the guidelines of the Helsinki Declaration on Human Experimentation.

### 2.2. Risk Stratification and Treatment

In accordance with ALL protocols, based on the immunophenotypic and molecular characteristics of the leukemic blasts and the treatment response, patients were classified into 3 risk groups: standard, intermediate, and high risk [23]. Considering treatment protocol intensity, the standard group includes patients with standard risk B-lineage ALL, the intermediate group includes medium risk B-lineage ALL and standard risk T-lineage ALL, and the high-risk group includes high risk B- and T-lineage ALL and chromosome Philadelphia-positive ALL. Total doses of chemotherapy, in particular the total cumulative doses of GCs (dexamethasone and prednisone), high-dose methotrexate, and PEG-asparaginase administered during induction phase, consolidation phase, and re-induction phase in the ALL-treatment protocols were recorded. To better calculate the effect of GCs, a dose equivalent of cortisone corresponding to prednisone plus dexamethasone received by each patient has been calculated according to conversion tables [24].

### 2.3. Clinical Data

Baseline characteristics of the ALL patients, including age, sex, treatment duration, and leukemia classification, were collected. Anthropometric parameters, including height, weight, and body mass index (BMI), were assessed. All anthropometric data were converted to age- and sex-matched standard deviation scores (SDS) using the national growth chart [25]. Data about recurrent fever, regarded as more than two episodes of fever (T > 38.5 °C) requiring hospitalization, bone pain presented at the onset of the disease, and the occurrence of osteonecrosis were recorded. Moreover, the maximum value of serum triglycerides raised during treatment was recorded.

### 2.4. Bone Metabolism Markers

We assessed biomarkers of bone formation as osteocalcin (bone matrix protein), P1NP (N-terminal propeptide of type I collagen), and bone alkaline phosphatase-bALP (osteoblast enzyme), as well as biomarkers of bone resorption as C-terminal telopeptide of type I collagen-CTX-I (marker of collagen breakdown) and TRACP5b (tartrate-resistant acid phosphatase 5b; osteoclast enzyme).

A venous blood sample was drawn from all participants at 08:00 a.m. after a 12 h fast. The serum samples from patients and controls were stored in aliquots at −20 °C for subsequent assay, with measurements made immediately after thawing. All serum bone markers were measured in the same assay run to avoid interassay variance.

Calcium and phosphorus concentrations were measured using the spectrophotometric method. Serum active intact parathyroid hormone (PTH) and 25(OH)-vitamin D were measured using immunological tests based on the principle of chemiluminescence using commercial kits (Liaison assay; DiaSorin, Stillwater, MN, USA).

Serum Osteocalcin was measured with enzyme immunoassay (IDS-iSYS N-MID^®^ Osteocalcin) (Catalog No. IS-2900) (Immunodiagnostic Systems LtD 10 Didct Way, Bol-don Business Park, Boldon, Tyne and Wear, UK, NE35 9PD), with an analytical sensitivity of 2 ng/mL and a linear range of 2–200 ng/mL. Serum P1NP was measured with enzyme immunoassay (IDS-iSYS Intact P1NP) (Catalog # IS-4000) (Immunodiagnostic Systems LtD 10 Didct Way, Boldon Business Park, Boldon, Tyne and Wear, NE35 9PD), with an analytical sensitivity of 2 ng/mL and a linear range of 2–230 ng/mL. Serum bALP was measured with enzyme immunoassay (IDS-iSYS Ostase^®^BAP) (Catalog No. IS-2800) (Immunodiagnostic Systems LtD 10 Didct Way, Boldon Business Park, Boldon, Tyne and Wear, NE35 9PD), with an analytical sensitivity of 1 µg/L and a linear range of 1–75 µg/L. Serum CTX was measured with enzyme immunoassay (IDS-iSYS CTX-I^®^ (CrossLaps^®^) (Catalog No. IS-3000) (Immunodiagnostic Systems LtD 10 Didct Way, Bol-don Business Park, Boldon, Tyne and Wear, NE35 9PD), with an analytical sensitivity of 0.033 ng/mL and a linear range of 0.033–6000 ng/mL. Serum TRAcP was measured with enzyme immunoassay (IDS-iSYS TRAcP 5b (BoneTRAP^®^) (Catalog No. IS-4100) (Immunodiagnostic Systems LtD 10 Didct Way, Bol-don Business Park, Boldon, Tyne and Wear, NE35 9PD), with an analytical sensitivity of 0.9 U/L and a linear range of 0.9–14.0 U/L. The dosage of CTX, Osteocalcin, TRAcP, bALP, and P1NP was performed with chemiluminescence assay using the TGSTA Technogenetics instrumentation (Techno-genetics, Milano-Italy).

All tests were performed in compliance with the manufacturer’s instructions and using suitable internal quality controls.

### 2.5. Bone Remodeling Cytokines Assessment

The analyses of RANKL, OPG, DKK-1, and sclerostin were performed using enzyme immunoassays (Immundiagnostik AG, Stubenwald-Allee 8a, D-64625 Bensheim for RANKL; Biomedica Medizinprodukle GmbH and Co KG, A-1210 Wien, Divischgasse for OPG; Bioclarma srl, Torino, Italy for DKK-1 and sclerostin). The dosage was performed with ELISA assay using the DSX^®^ TGSTA (Dynex Technologies, Inc., Chantilly, VA, USA. The specific characteristics are the following: RANKL: analytical sensitivity of 0.24 pmol/L and linear range of 0.24–4.800 pmol/L; OPG: analytical sensitivity of 0.07 pmol/L and linear range of 0.07–20 pmol/L; DKK-1: assay range 31.3–2000 pg/mL; and sclerostin: assay range 0–240 pmol/L.

### 2.6. Statistical Analys

Statistical analyses and data management were performed using SigmaPlot 12.0 for Windows and Chemometric Agile Tool (CAT) for univariate and multivariate analysis, respectively. Power analysis was performed for OPG marker difference between groups. It was indicated that a sample size of 49 controls and 39 ALL patients had 80% power to detect a mean difference (delta) in OPG values of 0.76 with a significance level (alpha) of 0.05 (two-tailed). All numerical variables were evaluated regarding normality in distribution both graphically (box plots) and statistically using the Shapiro–Wilk test, and regarding the homogeneity of variances verified using Levene’s test. Variables normally distributed were expressed as mean ± standard deviation, while variables with non-normal distribution were expressed as median and interquartile range (IQRs). Statistical comparisons among the controls and ALL patients of each bone marker were performed using a parametric Student’s test for normally distributed variables and a Mann–Whitney U test for variables that did not fulfill normality distribution. For comparisons between categorial variables (such as gender), Fisher’s exact test was used. Comparisons among more than two groups of patients were performed using parametric ANOVA or a Kruskal–Wallis test for variables with normal or non-normal distribution, respectively. Conventional univariate correlations among all bone markers and bone markers with GC dose were assessed using Spearman rank order correlation. The level of significance was set in all cases at *p* < 0.05. A multivariate statistical approach using principal component analysis (PCA) was performed to study possible hidden patterns of associations between bone markers in the patients and controls. PCA is a useful data mining process to investigate patterns of associations (covariance) within large numbers of variables, which cannot be revealed by classic inference statistics. Automatically derived factors, produced by standardized statistical procedure, explain a certain amount of overall variance of included variables, expressed as eigenvalues. Principal components are presented in descending order of overall variance explained. To obtain an interpretation of PCA, two plots are needed: A loading plot, which allowed us to understand the importance of each original variable in constructing the components and the type of correlation (positive or negative) between all of them. When variables are in the same area of the plot, they are positively correlated; variables located in opposite quarters are negatively correlated. The evaluation is limited to the components considered in the plot and is more significant if the explained variance of the component is relevant.

A score plot, which allowed us to evaluate the behavior of the data in the new orthogonal space defined by the principal components, highlighting similarities and differences among samples. The aim was to identify outliers, trends, and groupings or the occurrence of regularities and distributions among samples. Before starting PCA, a column autoscaling of data (Z-score values) was performed.

## 3. Results

### 3.1. Patient Population

The clinical characteristics of the ALL population are shown in Table 1. According to immunophenotype of blast cells, 32 subjects were diagnosed as B-lineage ALL, 5 as T-lineage ALL, and 2 were B-lineage ALL chromosome Philadelphia-positive (ALL Ph+). Patients were treated according to the AIEOP-BFM ALL 2017 and ESPhALL 2017 protocols. In total, 8 patients received ALL treatment according to high-risk protocols (5 patients with ALL B lineage, 1 ALL T lineage, 2 ALL Ph+), 18 intermediate risk, and 13 standard risk. One patient received cranial irradiation 12 Gy for central nervous system prophylaxis during the maintenance phase. None of the patients had received bone marrow transplantation at the time of recruitment.

Patients received a median cumulative dose of 1714.8 mg/m^2^ prednisone (range 420–2880 mg/m^2^), 356.9 mg/m^2^ dexamethasone (range 140–848.65 mg/m^2^), 20,472.5 mg/mq of cortisone equivalent (range 17,062.5–38,145.8 mg/m^2^), 18.2 g/m^2^ high-dose methotrexate (range 10–20 g/m^2^), and 9743.5 UI/m^2^ PEG-asparaginase (range 7500–20,000). As shown in Table 1, the majority of patients (32/39, 82%) received high-dose methotrexate at 20 g/m^2^, while 7/39 (8%) patients received a dose of 10 g/m^2^. Regarding PEG-asparaginase, the majority of patients (31/39, 79.5%) received a dose of 7500 UI/m^2^, while 8 patients in the high-risk group received higher doses: 12,500 UI/m^2^ (1/39, 2.5%), 15,000 UI/m^2^ (1/39, 2.5%), and 20,000 UI/m^2^ (6/39 15.4%). In total, 17 out of the 39 ALL patients (43.5%) presented recurrent fever during treatment. Bone pain was reported at the beginning of the disease in 7/39 (17.9%) patients. In total, 3 out of 39 (7.6%) patients presented severe osteonecrosis during the maintenance phase of therapy.

### 3.2. Bone Metabolism and Bone Remodeling Markers

Table 2 shows the bone metabolism and bone remodeling markers in the study population. Osteocalcin, CTX-I, and TRACP5b levels were significantly higher in the ALL population with respect to the controls (*p* = 0.004, *p* < 0.001, and *p* = 0.001, respectively). RANKL and OPG serum levels were significantly increased in ALL patients compared to the controls (*p* = 0.029 and *p* = 0.002, respectively) (Table 2, Figure 1). Moreover, bone turnover and bone remodeling markers were evaluated in the three risk groups: no significant differences were found comparing the different risk groups of ALL patients (*p* > 0.05).

To assess the effect of GC therapy on bone turnover markers and bone remodeling cytokines, the total amount of prednisone plus dexamethasone received was converted into a corticosteroid equivalent dose. No statistically significant correlation between cortisone dose and bone markers was found (*p* > 0.05).

Univariate marker correlations are presented in Table 3. Using the Spearman correlation coefficient, associations between bone remodeling and bone metabolism markers were investigated. OPG positively correlated with bALP (r = 0.40, *p* < 0.05); CTX positively correlated with osteocalcin (r = 0.55, *p* < 0.001) and P1NP (r = 0.50, *p* < 0.05); osteocalcin positively correlated with bALP and PTH (r = 0.38 and 0.43, respectively, *p* < 0.01) and correlated with TRACP5b and P1NP (r = 0.53, 0.69 and 0.51, respectively, *p* < 0.001); and bALP negatively correlated with age at diagnosis and at recruitment (r = −0.44 and −0.43, *p* < 0.01).

### 3.3. Principal Component Analysis of Biomarkers

PCA was performed to study all biomarkers in combined analysis to identify patterns and correlations. Considering nine of the main bone turnover and bone remodeling markers, a multivariate approach was used for the whole patient cohort (n = 88). The first step was to perform two preliminary tests to establish the possibility of running a PCA. The first test was the Kaiser–Meyer–Olkin test (KMO test), a measure of sampling adequacy to evaluate whether the variables considered were consistent for the use of a principal component analysis; the KMO test performed on our data set provided an acceptable value of 0.73 (its value ranging from 0 to 1 is consistent above 0.70). The second test was the Bartlett test, performed to test the null hypothesis of non-correlation between variables; it provided a statistically significant result with *p* < 0.001. Then, the PCA was used to evaluate the number of principal components which explain data variability. Two principal components were fixed as observed by the scree plot (Figure 2a). The first and the second principal component (PC1 and PC2) explained 33.6 and 13.5% of overall variance, respectively. The importance of each bone remodeling marker to build principal components was investigated. Interestingly, as shown in Figure 2b, the osteocalcin, P1NP, and CTX markers each explained more than 60% of variance, followed by bALP, TRACP5b, OPG, and DKK-1. RANKL and sclerostin showed very low relevance in both principal components. These results were confirmed by observing the loading plot in Figure 3. Remodeling markers with longer vectors had a strong weight in the principal components, while markers with short vectors did not have relevance in explaining variability and consequently differences between healthy and ALL patients. Moreover, from the loading plot, correlations between markers were observed. Two strong correlation patterns among the CTX, P1NP, osteocalcin, and TRACP5b markers and among bALP, OPG, and DKK-1 were found (Figure 3, blue and green circles). Considering the score plot (Figure 3), interestingly, the entire cohort of patients showed a clusterization between the controls and ALL patients (black and red circles) along the positive direction of PC1. As clusterization was observed along the first principal component, these results suggest that higher values of the P1NP, osteocalcin, CTX, and TRAcP markers could play a key role in the bone remodeling features of ALL patients with respect to the controls.

## 4. Discussion

In this study, we depicted the profile of serum bone cytokines and bone biomarkers in ALL children after the intensive phase of chemotherapy. Childhood, adolescent, and young adult ALL survivors show an increased risk of low BMD, osteonecrosis, and bone fragility fractures. The major treatment-related risk factors are represented by craniospinal radiotherapy, abdominal or pelvic irradiation, total body irradiation, and GCs [26]. However, the literature data evaluating the bone remodeling cytokines and bone metabolism biomarkers in ALL patients are lacking.

Bone health depends on the balance between bone resorption and bone deposition; in fact, most bone diseases reflect the imbalance between osteoclast and osteoblast activity. The role of RANKL/OPG and WNT-ß-catenin signaling pathways in the pathogenesis of several bone diseases has been well documented [27,28,29]. In addition, an impairment of both bone resorption and bone deposition has been observed in several acquired and congenital pediatric diseases [21,30].

In this study, we found higher levels of RANKL and OPG in ALL children compared to the controls, with a balance of RANKL/OPG ratio, whereas the levels of sclerostin and DKK-1, the main inhibitors of osteoblastogenesis, were similar in the two groups of subjects. These results suggest that in our cohort of ALL subjects, the intensive chemotherapy and GCs increased osteoclastic activity and, thereby, bone resorption.

GCs (prednisolone and dexamethasone) are keystone drugs in first-line ALL treatment and may be used throughout the entire treatment period. In particular, they have a key role during the initial phase of treatment, namely the induction of remission, with a strong cytotoxic effect on cancer cells by eliminating 99% of disease tissue, and during the re-induction phase, which counterbalances some adverse effects, such as those on BMD.

GCs enhance RANKL expression but decrease the production of its soluble decoy receptor, OPG, by osteoblastic cells, shifting the balance towards bone resorption [31,32]. In addition, GCs have a direct action on osteoclasts [33]. GIO occurs in two phases: an initial, rapid phase of bone loss due to excessive bone resorption and a second, slower phase due to decreased bone formation [31]. Recently, it has been demonstrated that GCs (dexamethasone, hydrocortisone, and prednisolone) reduce osteoclast number and size during the initial phases of RANKL-induced osteoclastogenesis [34]. In human osteoblastic cell lines, deflazacort has been demonstrated to be at least 10-fold less potent as an inhibitor of OPG compared to prednisolone and to stimulate RANKL expression less [35]. In addition, cancer cell-induced signals may alter the balance between RANKL and OPG, enhancing osteoclastogenesis and bone loss [36].

Based on our observations, we can hypothesize the role of CG treatment affecting RANKL expression in children treated for ALL, although no previous literature data on bone remodeling cytokines in ALL patients have been reported.

In our ALL cohort, we also found high levels of OPG which could be interpreted as an attempt to balance the high levels of RANKL.

In addition, it has been recently demonstrated that B-ALL cells isolated from patients at diagnosis can determine bone destruction through RANK-RANKL axis [37]. The standard of care for B-ALL patients is intensified multi-agent chemotherapy, including GCs, which independently represents a significant risk of osteoporosis and osteonecrosis. The finding of high RANKL levels provided evidence that RANKL expressed on B-ALL cells can be a critical mediator of bone loss in these patients. In our cohort, 32 subjects out of the 39 ALL enrolled were diagnosed as B-lineage ALL. According to the results of our study, depicting for the first time the scenario of bone markers in pediatric ALL after intensive chemotherapy, we cannot exclude that a causative role relays in the disease itself in addition to being a consequence of the delivered treatment. However, we should consider that all the patients are in molecular complete remission of the disease at the moment of recruitment to this study; therefore, due to the absence of blast cells, the direct effect of the disease itself is likely negligible. At the moment, this is a limitation of our study, and it would be interesting, as a further step of our study, to clarify the role of leukemia cells themselves through a prospective evaluation of bone remodeling cytokines and bone biomarkers from the onset of leukemia, during treatment, and until the end of therapy. Indeed, previous studies demonstrated that at time of diagnosis, the incidence of vertebral fractures ranges from 1.5% to 16%, and a younger age and lower weight represent risk factors [10,11,38,39].

To confirm the shift of bone remodeling in our cohort of ALL patients towards bone resorption, there is also the finding of higher levels of CTX and TRACP5b in ALL subjects than in the controls. CTX is released into the circulation when collagen is degraded during bone resorption, and it is excreted in urine [40]. Increased serum levels of CTX have been observed 24 h after prednisolone treatment [41]. TRACP-5b is secreted by osteoclasts as an active enzyme, and its serum concentration reflects bone resorption; however, it must be noted that it reflects osteoclast number rather than osteoclast activity [40]. In addition, we found higher levels of osteocalcin in ALL patients than in the controls, as a marker of bone formation. It has been observed that ALL children who received dexamethasone during the third period of intensification chemotherapy showed low ALP and collagen markers, but partial recovery was also observed after the discontinuation of dexamethasone [42].

Our results demonstrated no statistically significant correlation between bone biomarkers, bone remodeling cytokines, and variables related to leukemia, such as immunophenotype, risk assignment, or treatment provided. We did not demonstrate any effect of different doses of CG or chemotherapy on bone biomarker profile. It might be that the exposure to intensive phases of treatment for ALL affect bone remodeling biomarkers, which are a sensitive test at the “preclinical level” of bone health, in a deep and generalized fashion in which we could not catch slight differences. Further studies on a larger sample size could give more information.

Osteonecrosis represents one of the most described adverse effects in ALL children and adolescents with a long-term negative impact on quality of life [13]. In our study, population osteonecrosis is reported in 3/39 (7.6%) children, which is in line with previous data studies. We did not find any correlation between osteonecrosis and bone remodeling markers in our ALL cohort.

## 5. Conclusions

In this study, we took a screenshot of biochemical markers of bone metabolism in ALL patients, as bone is a target for the toxic effects of chemotherapy. The reduction of BMD is associated with ALL, and it has been demonstrated that the altered bone remodeling due to the RANK/RANKL/OPG pathway is responsible for bone loss in several pediatric diseases.

The chemotherapy protocols used for the treatment of ALL result in the stimulation of bone resorption which is not counterbalanced by adequate bone formation.

Leukemia per se could be a predisposing factor to low BMD in these patients, as observed in a few previous studies. A prospective evaluation of bone remodeling markers from diagnosis to the end of treatment is needed in a future research plan.

In conclusion, our study describes for the first time a signature of bone health in ALL patients, which might offer a better understanding of the mechanism leading to osteopenia/osteoporosis and fractures in these patients. Innovative therapies, such as the use antiresorptive drugs and monoclonal antibodies for the treatment of osteopenia/osteoporosis, are being developed also in pediatric patients and could then also be considered in ALL patients.

## Figures and Tables

**Figure 1 cancers-15-02554-f001:**
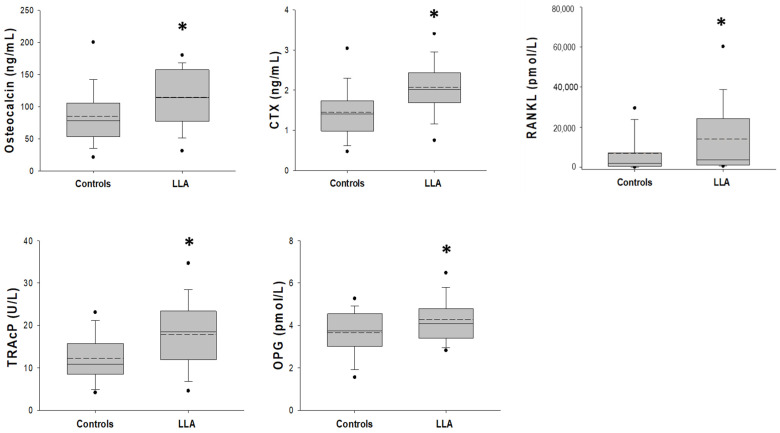
Distribution of bone remodeling markers values, osteocalcin, CTX, RANKL, TRAcP and OPG, measured for healthy and LLA patients. The mean and median values are marked by dotted and black lines, respectively. The superscript * in each box plot indicates statistical difference between the two groups of patients (*p* < 0.05).

**Figure 2 cancers-15-02554-f002:**
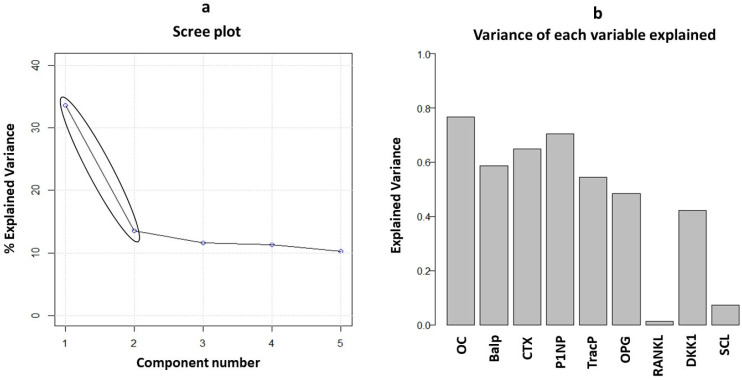
(**a**,**b**) Scree plot: percentage of explained variance vs. component number of principal component of the model (in general, the component number before the inflection point is retained). Variance of each variable explained: the weight of each variable to the two principal components selected in the PCA model.

**Figure 3 cancers-15-02554-f003:**
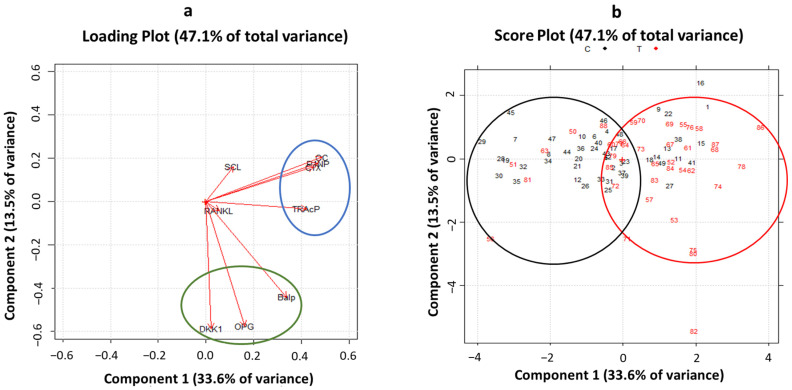
Loading and score plots of PCA (principal components 1 and 2) of bone remodeling markers. In the loading plot (**a**), blue and green circles represent correlation patterns between variables. In the score plot (**b**), black and red circles (and respective numbers) represent clusterization of the healthy and ALL patients, respectively. DKK1: Dickkopf-related protein 1, RANKL: soluble receptor activator of nuclear factor kappa-B ligand, OPG: osteoprotegerin, CTX: C-terminal telopeptide cross-links of type I collagen, bALP: total alkaline phosphatase, TRAcP5b: tartrate-resistant acid phosphatase, P1NP: procollagen type I N-terminal propeptide, OC: Osteocalcin, and SCL: sclerostin.

**Table 1 cancers-15-02554-t001:** Clinical characteristics of ALL patients.

Clinical Characteristics	Patients (n = 39)
Gender (male/female)	21/18
Age at recruitment—years	7.64 ± 4.47
Age at diagnosis—years	6.24 ± 4.25
Height (SDS)	−0.75 ± 1.26
Weight (SDS)	−0.14 ± 1.18
BMI (SDS)	0.49 ± 0.91
ALL phenotype: B-lineage	32
T-lineage	5
B-lineage t (9;22)	2
RISK GROUP: standard risk	13
Intermediate risk	18
High risk	8
Bone pain yes/no	7/32
Recurrent fever yes/no	17/22
Triglycerides mg/dL	186.03 ± 176.94
Corticosteroid dose mg/m^2^	20,472.53 ± 7023.68
HD-MTX (10 g m^−2^/20 g m^−2^)	7/32
PEG-ASP UI/m^2^ 7500	31
12,500	1
15,000	1
20,000	6

SDS: standard deviation score; HD-MTX: high-dose methotrexate; PEG-ASP: PEG-asparaginase.

**Table 2 cancers-15-02554-t002:** Bone metabolism and bone remodeling markers in the study population.

	Controls (n = 49)	Patients (n = 39)	*p* Value
Osteocalcin (ng/mL)	85.2 ± 46.4	114.4 ± 46.8	0.004 ^a^
P1NP (ng/mL)	692 (532∼948)	751 (700∼852)	0.305 ^b^
bALP (µg/L)	73.7 (55.9∼97.6)	77.6 (54.6∼115.5)	0.389 ^b^
CTX (ng/mL)	1.4 (1.0∼1.7)	2.0 (1.7∼2.4)	<0.001 ^b^
TRAcP5b (U/L)	10.9 (8.4∼16.4)	18.5 (11.8∼23.8)	0.001 ^b^
RANKL (pmol/L)	1954 (350∼7258)	3759 (1073∼24,976)	0.029 ^b^
OPG (pmol/L)	3.7 ± 1.2	4.3 ± 1.2	0.020 ^a^
RANKL/OPG ratio	581 (89∼2345)	762 (238∼4226)	0.098 ^b^
DKK1 (pg/mL)	5537 (4335∼6287)	5513 (4466∼7168)	0.480 ^b^
Sclerostin (pmol/L)	17.8 (15.3∼19.9)	16.9 (14.2∼19.3)	0.252 ^b^

Data are presented as actual numbers (%) for proportions, mean ± SD for normally distributed variables, and median with interquartile range for non-normally distributed variables. P1NP: procollagen type I N-terminal propeptide; bALP: bone-specific total alkaline phosphatase; CTX: C-terminal telopeptide cross-links of type I collagen; TRAcP5b: tartrate-resistant acid phosphatase; RANKL: soluble receptor activator of nuclear factor kappa-B ligand; OPG: osteoprotegerin, DKK1: Dickkopf-related protein. ^a^ Student *t*-test; ^b^ Mann–Whitney U test.

**Table 3 cancers-15-02554-t003:** Marker correlations in ALL patients using Spearman rank order correlation.

	DKK1	Sclerostin	RANKL	OPG	25OH Vit D	CTX	OC	bALP	TRAcP	P1NP	Ca	P	PTH	IL6	Age at Diagnosis	Age at Recruitment	Weight	Height	BMI
**DKK-1**	1																		
**Sclerostin**	0.18	1																	
**RANKL**	0.15	0.01	1																
**OPG**	0.10	0.03	−0.04	1															
**25OH Vit D**	−0.23	−0.04	0.07	0.11	1														
**CTX**	−0.01	−0.02	0.15	0.06	−0.10	1													
**OC**	−0.05	0.05	−0.06	0.12	−0.19	0.55 ***	1												
**bALP**	−0.10	−0.17	−0.06	0.40 *	−0.02	0.24	0.38 *	1											
**TRAcP**	−0.14	−0.06	0.15	0.14	−0.09	0.29	0.53 ***	0.33	1										
**P1NP**	0.01	0.16	0.08	−0.09	−0.16	0.50 *	0.69 ***	0.30	0.63 ***	1									
**Ca**	−0.06	−0.20	−0.22	0.27	−0.08	−0.16	−0.05	0.04	0.05	0.032	1								
**P**	−0.06	−0.06	−0.02	0.12	−0.04	0.50 *	0.51 ***	0.14	0.33 *	0.46 **	0.24	1							
**PTH**	−0.04	−0.04	−0.12	−0.05	−0.32	0.10	0.43 **	−0.03	0.19	0.33 *	−0.12	−0.01	1						
**Age at diagnosis**	−0.15	0.24	0.17	−0.29	−0.15	0.05	−0.12	−0.44 **	0.06	−0.13	−0.21	−0.20	0.04	−0.04	1				
**Age at recruitment**	−0.33 *	0.23	0.17	−0.32	−0.11	−0.04	−0.16	−0.43 **	0.03	−0.16	−0.10	−0.16	0.05	−0.01	0.92	1			
**Weight**	−0.08	0.15	−0.28	0.02	−0.04	−0.20	−0.04	−0.10	0.12	0.09	−0.10	−0.08	−0.10	−0.15	0.09	0.01	1		
**Height**	−0.03	0.04	−0.06	−0.23	−0.26	−0.06	−0.09	−0.27	0.09	0.02	−0.13	−0.20	−0.07	−0.33	0.40	0.28	0.73	1	
**BMI**	0.002	0.11	−0.28	0.32 *	0.16	−0.33 *	−0.02	0.03	−0.01	0.05	0.08	0.03	−0.10	−0.06	−0.33	−0.34	0.70	0.10	1

DKK-1: Dickkopf-related protein 1; RANKL: soluble receptor activator of nuclear factor kappa-B ligand; OPG: osteoprotegerin; 25OH Vit D: 25 hydroxy vitamin D; CTX: C-terminal telopeptide cross-links of type I collagen; bALP: total alkaline phosphatase; TRAcP5b: tartrate-resistant acid phosphatase; P1NP: procollagen type I N-terminal propeptide; Ca: calcium; P: phosphorus, PTH: parathormone. Weight (W), Height (H), and Body mass index (BMI) are expressed as SDS. The superscripts * above each value indicate statistically significant correlation between the two markers considered. Superscript * indicates a statistical significance with *p* ≤ 0.05; superscript ** indicates a statistical significance with *p* ≤ 0.01; superscript *** indicates a statistical significance with *p* ≤ 0.001

## Data Availability

The data presented in this study are available on request from the corresponding author.

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
