# Peer review of "Bone Remodeling Markers in Children with Acute Lymphoblastic Leukemia after Intensive Chemotherapy: The Screenshot of a Biochemical Signature"

_cancers, 2023, doi:10.3390/cancers15092554_

Round 1

Reviewer 1 Report

The manuscript is organized in a logical manner. The aim of this study was to investigate the effects of intensive chemotherapy and glucocorticoid (GC) treatment on bone remodeling markers in children with acute lymphoblastic leukemia (ALL). The statistical analysis has been performed appropriately. The interpretation of the results is also clearly presented. The authors concluded that children with ALL showed an enhanced bone resorption. This study is very important for identification ALL patients who are most at risk of developing bone damage and who need preventive interventions. The manuscript is acceptable after minor revision: methodology of osteocalcin, P1NP, bALP, TRACP5b, and CTX-I assessment should be describe briefly.

Author Response

The manuscript is organized in a logical manner. The aim of this study was to investigate the effects of intensive chemotherapy and glucocorticoid (GC) treatment on bone remodeling markers in children with acute lymphoblastic leukemia (ALL). The statistical analysis has been performed appropriately. The interpretation of the results is also clearly presented. The authors concluded that children with ALL showed an enhanced bone resorption. This study is very important for identification ALL patients who are most at risk of developing bone damage and who need preventive interventions. The manuscript is acceptable after minor revision: methodology of osteocalcin, P1NP, bALP, TRACP5b, and CTX-I assessment should be described briefly.

Answer: we really thank the reviewer for his/her appreciation. We described the methodology of osteocalcin, P1NP, bALP, TRACP5b, and CTX-I assessment in the methods section.

Reviewer 2 Report

This is potentially a very interesting area of research. Now that childhood acute lymphoblastic leukaemia (ALL) cure rates are above 80% meaning an increase in the number of survivors, it is very important to describe and prevent therapy-related long term side effects, such as bone alterations. However, the study proposed by Muggeo et al. has some limitations. Here are some comments:

-         The title “Bone remodeling markers in children with acute lymphoblastic leukemia after intensive chemotherapy: the screenshot of a biochemical signature” does not reflect the proper content of the manuscript. The authors compare ALL patients who have already received chemotherapy treatments with healthy subjects (control group) but the data from a group of patients at diagnosis is missing, so how can be assumed the alterations described on different bone remodeling markers are produced by therapy and not by the disease? As authors mention in the introduction it has already been shown that these patients have less bone mineral density and osteoporosis at diagnosis (ref 10,11). Thus, comparison between ALL patients before and after the treatment is of critical importance and would provide new data to the field but is not included in this study.

-         Different risk groups and therefore different treatment intensity protocols are included in the study. They find no significant differences between different risk groups and no correlation between cortisone dose and bone markers. These results are not properly discussed and if the problem, as authors say, is the sample size, then patient’s cohort should be increased or more homogenous in order to have more reliable results.

Overall, I encourage the authors to continue with this topic and build a more complete study around this initial data.

Author Response

This is potentially a very interesting area of research. Now that childhood acute lymphoblastic leukaemia (ALL) cure rates are above 80% meaning an increase in the number of survivors, it is very important to describe and prevent therapy-related long term side effects, such as bone alterations. However, the study proposed by Muggeo et al. has some limitations. Here are some comments:

-         The title “Bone remodeling markers in children with acute lymphoblastic leukemia after intensive chemotherapy: the screenshot of a biochemical signature” does not reflect the proper content of the manuscript. The authors compare ALL patients who have already received chemotherapy treatments with healthy subjects (control group) but the data from a group of patients at diagnosis is missing, so how can be assumed the alterations described on different bone remodeling markers are produced by therapy and not by the disease?

As authors mention in the introduction it has already been shown that these patients have less bone mineral density and osteoporosis at diagnosis (ref 10,11). Thus, comparison between ALL patients before and after the treatment is of critical importance and would provide new data to the field but is not included in this study.

Answer: we really thank the reviewer for his/her suggestion. In this study, we aimed to investigate in a cohort of ALL children the effects of intensive phases of chemotherapy on bone remodeling, and to characterize the unknown biochemical signature of bone status in these patients. We really agree that the leukemia per se could be a predisposing factor for low BMD in these patients as observed in few studies (see reference 10,11). However, in this study, as first line of our research, we focused on the status of bone remodeling after intensive treatment in children in complete remission of leukemia, while we are prospectively collecting data from the diagnosis to the end of treatment for a future study. Different cooperative groups have reported on osteonecrosis after intensive chemotherapy in children and young patients treated for ALL, arguing this as toxic effect from chemotherapy.   We cannot exclude the effect of leukemia blasts on bone health added to the effect of treatment, anyway for the first time, at the best of our knowledge, we depicted bone remodeling markers after intensive chemotherapy in children with ALL. This concept has been better expressed in the discussion stressing the limitation of the study and we added this observation in the conclusion.

- Different risk groups and therefore different treatment intensity protocols are included in the study. They find no significant differences between different risk groups and no correlation between cortisone dose and bone markers. These results are not properly discussed and if the problem, as authors say, is the sample size, then patient’s cohort should be increased or more homogenous in order to have more reliable results. Overall, I encourage the authors to continue with this topic and build a more complete study around this initial data.

Answer: thank you very much for this comment. Differently from osteonecrosis and BMD for which the literature has a plenty of studies, for bone remodeling biomarkers we did not find any studies to compare our experience. Our study could represent a first step in this field, reporting what happens from the biological point of view in the bone of children treated for ALL. We can hypotesize that bone biomarkers are a more sensitive test for bone health as compared to MR imaging, and therefore intensive treatment in the course of ALL affects them in a deep and generalized fashion. Such preliminary results will be confirmed by the prospective study. We also discussed the lack of correlation with cortisone doses.

Reviewer 3 Report

In this study, Paola et al., investigated the effect of intensive chemotherapy on the bone remodeling of ALL patients. I am not surprised to see that chemotherapy affects the bone remodeling process. In Fig.1, markers associated with bone resorption or formation are significantly increased in ALL patients.  Therefore, any conclusion based on the expression levels of markers would be difficult to make. Additionally, I don’t see any cytogenetic data for ALL cohorts, stratifying patients based on genetic alternation (high /low risk) and then correlating with the expression levels on bone remodeling markers would have strengthened this manuscript. 

Author Response

In this study, Paola et al., investigated the effect of intensive chemotherapy on the bone remodeling of ALL patients. I am not surprised to see that chemotherapy affects the bone remodeling process. In Fig.1, markers associated with bone resorption or formation are significantly increased in ALL patients.  Therefore, any conclusion based on the expression levels of markers would be difficult to make.

Answer: in our cohort of ALL subjects, we observed higher levels of bone resorption markers such as CTX-I, TRACP5b and RANKL compared to controls, suggesting a therapy-induced imbalance between bone resorption and bone deposition. Moreover, the increase in osteocalcin and OPG levels represent an attempt to counterbalance this enhanced bone reabsorption. The osteodeposition would not seem to be compromised as we found overlapping levels of DKK1 sclerostin and P1NP between patients and controls. Such picture has been described for the first time, at the best of our knowledge.

-Additionally, I don’t see any cytogenetic data for ALL cohorts, stratifying patients based on genetic alternation (high /low risk) and then correlating with the expression levels on bone remodeling markers would have strengthened this manuscript. 

Answer: we really thank the reviewer for this observation. Molecular and cytogenetic data are avalaible for  ALL patients at the onset of the disease (they are characterized as follows: ETV6/RUNXL 7 patients, CDKN2A/B 3 patients, BCR/ABL 2 patients, PAX5 2 patients,CLL1 2 patients, no genetic alterations 23 patients), however in modern ALL treatment protocols they are part of a dinamic risk stratification which takes into account more parameters including the response to treatment as measured by molecular minimal residual disease (MRD). Therefore, the prognostic impact of cytogenetic markers should be assessed in combination with MRD values and clinical characteristics of patients. The risk stratification reported in our study population is the result of immunophenotyping, genetic characteristics as well as minimal residual disease measurement.

Moreover, we should consider that our study is focusing on a population of children in remission of the disease. In such condition it is likely that we are measuring the effect of treatment rather than the effect of leukemic blasts on bone remodeling. As second step we are carring on a study on bone remodeling in ALL from the onset of the disease until stopping chemotherapy. This latter concept has been also added in the discussion.

Round 2

Reviewer 2 Report

Dear authors,

The present manuscript has improved. All limitations are no know more clearly exposed and adequately discussed. However, I would like to suggest an improvement in figures quality.

Best regards

Author Response

Answer: we improved the quality of figures.

Reviewer 3 Report

I am not convinced by the author’s arguments, they need to be supported by scientific data. While bone remodeling markers are increased in ALL patients, any conclusions based on these markers must be supported by the BMD data, rather than assuming that “osteodeposition would not seem to be compromised”. I would like to see a scientific answer supported by the data regarding the overall conclusion of the study. Additionally, the statement “This study was focused on a population of children in remission, and you are carrying on a study from the onset of the disease until stopping chemotherapy” prompts me to ask whether there is data available on bone remodeling prior to treatment. Such information would help to understand which bone remodeling markers are most impacted by the therapy. 

Author Response

Answer: we thank the referee to allow us to clarify this issue. Bone mass can be assessed through different methods that include the analysis of biochemical and/or instrumental parameters. Biochemical markers of bone remodeling are classified according to their function, as markers of bone formation such as bone-specific alkaline phosphatase, osteocalcin, and N-terminal propeptide of type I procollagen (P1NP), and markers of bone resorption involved in bone remodeling, such as terminal telopeptide of type I collagen (CTX). In addition, bone remodeling is influenced by two pathways: the RANK/RANKL/OPG and Wnt/β-catenin pathways, which play a key role in the control of osteoclastogenesis and osteoblastogenesis, respectively. The main instrumental tool for the evaluation of BMD, is dual X-ray absorptiometry (DEXA). In recent decades, biochemical markers of bone metabolism helped the clinicians to characterize alterations of skeletal metabolism in several pediatric diseases and for monitoring the effects of pharmacological therapies with adverse effects on bone such as glucocorticoids. The authors have published numerous studies on the importance of using these biomarkers in understanding the mechanisms of altered bone health in various pediatric pathologies (Chiarito M, Piacente L, Chaoul N, Pontrelli P, D'Amato G, Grandone A, Russo G, Street ME, Wasniewska MG, Brunetti G, Faienza MF. Role of Wnt-signaling inhibitors DKK-1 and sclerostin in bone fragility associated with Turner syndrome. J Endocrinol Invest. 2022 Jun;45(6):1255-1263. doi: 10.1007/s40618-022-01760-3; Giordano P, Urbano F, Lassandro G, Faienza MF. Mechanisms of Bone Impairment in Sickle Bone Disease. Int J Environ Res Public Health. 2021 Feb 13;18(4):1832. doi: 10.3390/ijerph18041832; Brunetti G, D'Amato G, Chiarito M, Tullo A, Colaianni G, Colucci S, Grano M, Faienza MF. An update on the role of RANKL-RANK/osteoprotegerin and WNT-ß-catenin signaling pathways in pediatric diseases. World J Pediatr. 2019 Feb;15(1):4-11. doi: 10.1007/s12519-018-0198-7; Brunetti G, Grugni G, Piacente L, Delvecchio M, Ventura A, Giordano P, Grano M, D'Amato G, Laforgia D, Crinò A, Faienza MF. Analysis of Circulating Mediators of Bone Remodeling in Prader-Willi Syndrome. Calcif Tissue Int. 2018 Jun;102(6):635-643. doi: 10.1007/s00223-017-0376-y; Brunetti G, D'Amato G, De Santis S, Grano M, Faienza MF. Mechanisms of altered bone remodeling in children with type 1 diabetes. World J Diabetes. 2021 Jul 15;12(7):997-1009. doi: 10.4239/wjd.v12.i7.997). It is clear that the use of these markers does not allow to prevent or identify the risk of fractures since the definition of osteoporosis in children is based on the assessment of BMD by DXA (Gordon CM, Leonard MB, Zemel BS; International Society for Clinical Densitometry. 2013 Pediatric Position Development Conference: executive summary and reflections. J Clin Densitom. 2014 Apr-Jun;17(2):219-24. doi: 10.1016/j.jocd.2014.01.007).

In this study we aimed to evaluate bone metabolism and bone remodeling biomarkers after intensive phases of chemotherapy in a cohort of ALL children, as no previous data are available in literature in these subjects. Our results demonstrated that in this cohort of ALL subjects there is an increase in bone resorption (high RANKL levels) with an attempt to counteract the activation of osteoclastogenesis (high OPG levels). As DKK1 and Sclerostin are in normal levels, we could hypotesize that the effects of GCs and intensive chemotherapy in these subjects did not alter the osteodeposition. Anyway, we slighty modified some sentences in abstract and aims.

-Additionally, the statement “This study was focused on a population of children in remission, and you are carrying on a study from the onset of the disease until stopping chemotherapy” prompts me to ask whether there is data available on bone remodeling prior to treatment. Such information would help to understand which bone remodeling markers are most impacted by the therapy. 

Answer: we are collecting the data on a prospective evaluation of these markers on ALL subjects but at the moment we are not able to add these informations in this paper.

Round 3

Reviewer 3 Report

No additional comments.